# CD74 and CD44 Expression on CTCs in Cancer Patients with Brain Metastasis

**DOI:** 10.3390/ijms22136993

**Published:** 2021-06-29

**Authors:** Desiree Loreth, Moritz Schuette, Jenny Zinke, Malte Mohme, Andras Piffko, Svenja Schneegans, Julia Stadler, Melanie Janning, Sonja Loges, Simon A. Joosse, Katrin Lamszus, Manfred Westphal, Volkmar Müller, Markus Glatzel, Jakob Matschke, Christoffer Gebhardt, Stefan W. Schneider, Iwona Belczacka, Beate Volkmer, Rüdiger Greinert, Marie-Laure Yaspo, Patrick N. Harter, Klaus Pantel, Harriet Wikman

**Affiliations:** 1Department of Tumor Biology, University Medical Center Hamburg-Eppendorf, 20246 Hamburg, Germany; d.loreth@uke.de (D.L.); s.schneegans@uke.de (S.S.); s.joosse@uke.de (S.A.J.); pantel@uke.de (K.P.); 2Alacris Theranostics GmbH, Max-Planck-Straße 3, 12489 Berlin, Germany; m.schuette@alacris.de; 3Institute of Neurology (Edinger Institute), Goethe University, 60528 Frankfurt am Main, Germany; jenny.zinke@gmx.net (J.Z.); patrick.harter@kgu.de (P.N.H.); 4German Cancer Consortium (DKTK), Partner Site Frankfurt/Mainz, 60528 Frankfurt am Main, Germany; 5German Cancer Research Center (DKFZ), 69120 Heidelberg, Germany; 6Frankfurt Cancer Institute (FCI), 60528 Frankfurt am Main, Germany; 7Department of Neurosurgery, University Medical Center Hamburg-Eppendorf, 20246 Hamburg, Germany; m.mohme@uke.de (M.M.); a.piffko@uke.de (A.P.); lamszus@uke.uni-hamburg.de (K.L.); westphal@uke.de (M.W.); 8Department of Dermatology and Venerology, University Hospital Hamburg-Eppendorf, 20246 Hamburg, Germany; j.stadler@uke.de (J.S.); ch.gebhardt@uke.de (C.G.); st.schneider@uke.de (S.W.S.); 9Division of Personalized Medical Oncology (A420), German Cancer Research Center (DKFZ), 69120 Heidelberg, Germany; melanie.janning@medma.uni-heidelberg.de (M.J.); s.loges@dkfz.de (S.L.); 10Department of Personalized Oncology, University Hospital Mannheim, Medical Faculty Mannheim, University of Heidelberg, 68167 Mannheim, Germany; 11 Department of Oncology, Hematology and Bone Marrow Transplantation with Section Pneumology, Hubertus Wald Comprehensive Cancer Center Hamburg, University Medical Center Hamburg-Eppendorf, 20246 Hamburg, Germany; 12Department of Gynecology, University Medical Center Hamburg-Eppendorf, 20246 Hamburg, Germany; v.mueller@uke.de; 13Department of Neuropathology, University Medical Centre Hamburg-Eppendorf, 20246 Hamburg, Germany; m.glatzel@uke.de (M.G.); matschke@uke.de (J.M.); 14Otto Warburg Laboratory Gene Regulation and Systems Biology of Cancer, Max Planck Institute for Molecular Genetics, 14195 Berlin, Germany; i.belczacka@gmail.com (I.B.); yaspo@molgen.mpg.de (M.-L.Y.); 15Association of Dermatological Prevention, Germany & Centre of Dermatology, Elbe Clinics, 21614 Buxtehude, Germany; beate.volkmer@elbekliniken.de (B.V.); ruediger.greinert@elbekliniken.de (R.G.)

**Keywords:** brain metastasis, circulating tumor cells, breast cancer, NSCLC, melanoma

## Abstract

Up to 40% of advance lung, melanoma and breast cancer patients suffer from brain metastases (BM) with increasing incidence. Here, we assessed whether circulating tumor cells (CTCs) in peripheral blood can serve as a disease surrogate, focusing on CD44 and CD74 expression as prognostic markers for BM. We show that a size-based microfluidic approach in combination with a semi-automated cell recognition system are well suited for CTC detection in BM patients and allow further characterization of tumor cells potentially derived from BM. CTCs were found in 50% (7/14) of breast cancer, 50% (9/18) of non-small cell lung cancer (NSCLC) and 36% (4/11) of melanoma patients. The next-generation sequencing (NGS) analysis of nine single CTCs from one breast cancer patient revealed three different CNV profile groups as well as a resistance causing ERS1 mutation. CD44 and CD74 were expressed on most CTCs and their expression was strongly correlated, whereas matched breast cancer BM tissues were much less frequently expressing CD44 and CD74 (negative in 46% and 54%, respectively). Thus, plasticity of CD44 and CD74 expression during trafficking of CTCs in the circulation might be the result of adaptation strategies.

## 1. Introduction

The incidence of brain metastases (BM) in many tumor entities has been increasing in the last few decades [1]. This is attributed to improvements in systemic therapy that are obviously lacking efficacy in the brain. BM are the most frequent type of brain tumors, arising mainly from primary tumors in the lung, the breast and the skin, with frequencies up to 40% [1,2]. Poor overall survival and deficits in cognitive and sensory functions are typical characteristics of BM [3,4,5], leading in many patients to severe impairments of routine daily life activities. Therefore, a better understanding of the underlying mechanisms for the development of BM is of high clinical relevance.

Metastasis arises from circulating tumor cells (CTCs). Analyses of these cells in peripheral blood are considered as an alternative to invasive tumor biopsies to detect and monitor tumors in patients [6,7]. We, and others, have shown that CTCs from breast and lung cancer patients with BM have a more mesenchymal state and are, thus, poorly detected with the commonly used EpCAM-dependent isolation systems [8,9,10,11]. Therefore, microfluidic devices using size-based CTC enrichment platforms might be more suitable for the detection of CTCs in BM patients. Further characterization of these CTCs might help in identifying patients prone to develop BM.

CD74 was first identified in immune cells, including microglia and, thus, plays an important role in immune surveillance [12]. It is also expressed on many tumors and is involved in the regulation of endosomal trafficking, cell migration and cellular signaling [13,14]. CD44 is a well-studied cell surface adhesion receptor and a stem cell marker [15]. CD44 has been shown to be expressed on metastasis-initiating CTCs and CTC clusters in breast cancer, thereby promoting tumorigenesis and metastasis [16,17]. In addition, CD44 and CD74 have been described to have important cooperative functions. In B-cells, the macrophage migration inhibitory factor (MIF) binding to a complex of CD74 and CD44 represents a major B-cell survival regulatory pathway [18]. Moreover, in primary breast cancer, a direct interaction between CD44 and CD74 has been described [12]. Both proteins have recently been shown to also be associated with BM development in breast cancer, NSCLC, and melanoma [19,20,21,22,23,24,25]. In the case of CD74 expression, the cellular context and micro milieu seem to be of importance, as its expression on immune cells (mostly microglia) and brain metastatic tumor cells has been described as a biomarker for positive patient survival [8,23,26]. Thus, these two proteins could be important prognostic liquid biopsy markers for brain malignancies. The aims of this study were to validate the size- and microfluidics-based Parsortix^®^ method for CTC detection in BM patients, and then analyze whether CD74 and CD44 expression on CTCs could be used as a stratification marker for BM.

## 2. Results

### 2.1. Size-Based Enrichment of CTCs in Breast Cancer, NSCLC and Melanoma Patients

CTCs from 7.5 mL whole blood of brain metastatic breast cancer, NSCLC, and melanoma patients were enriched using Parsortix^®^ and detected with a recently established protocol for semi-automatic CTC detection and quantification using the Xcyto^®^ Quantitative cell imager [13]. After enrichment, CTCs were stained for either pan-keratins for breast cancer and NSCLC or PMEL17/Melan-A for melanoma samples, DAPI for nuclear identification and for the leucocyte exclusion marker CD45 (Appendix A). Different types of cells with the above mentioned traits were identified: the large majority of the detected cells had a normal roundish shape with a diameter between 8 and 20 µm. These cells were defined as classical CTCs. Two large CTC clusters containing five and six cells could be identified in one melanoma patient and in one breast cancer patient a cluster of two cells was found. In addition, cells with a diameter more than 25 µm and/or several nuclei (hereinafter referred to as “large cells and multinucleated cells”) could also be detected in 2/14 breast cancer and 2/18 NSCLC samples (Figure 1A). Classical CTCs were found in half of breast cancer patients (7/14, CTC median 5, range 3–16) and also in half of NSCLC patients (9/18, median 3, range 1–19), whereas in melanoma 36% (4/11, median 6.5, range 1–12) had ≥one CTC per 7.5 mL blood (Figure 1B). Applying a cut-off of ≥two CTCs per 7.5 mL blood showed a positivity rate of 50% (7/14) for breast cancer, 33% (6/18) for NSCLC and 27% (3/11) for melanoma BM patients. The CTC-positive patients per tumor entity were further subdivided as only classical/normal CTCs, normal CTCs + large cells, normal CTCs and multinucleated as well as normal CTCs and clusters. This subdivision is displayed in Figure 1B. No patient with only large cells, only multinucleated or only clusters could be observed. Comparing the three entities, no significant difference in the median CTC count per patient could be detected (Figure 1C).

Among the five breast cancer patients with only-brain metastases (oligo-brain), 40% had CTCs whereas 53% (8/18) of NSCLC and 9% (1/11) of melanoma patients with oligo-metastatic disease had CTCs. In the patients with also other non-cerebral metastases, 55% of breast cancer, 33% of NSCLC and 40% of melanoma patients had detectable CTCs (Figure 1D). The median CTC number in breast cancer patients with an oligo-metastatic setting was found to be lower compared to patients with other peripheral metastases (3.5 range 3–4 versus 5.0 range 4–15). However, this difference did not reach significance (*p* = 0.160; Figure 1E). For NSCLC, the median number of CTCs in the oligo-brain metastatic state was found to be slightly higher (three range 1–12 versus one (only 1 CTC), *p* = 0.419; Figure 1E). As there was only one melanoma patient with oligo–brain metastatic disease, no conclusion can be drawn for this tumor entity (Figure 1E, *p* = 0.400).

When the CTC status was correlated with clinical parameters, a significant association was found for melanoma CTCs with the occurrence of concurrent liver metastases (*p* = 0.003) and a poorer overall survival (cut-off ≥1 CTCs per 7.5 mL blood, *p* = 0.049) (Table 1 and Appendix A–C). Survival analyses did not reveal an association between the CTC status and overall survival in breast cancer (Appendix A; cut-off ≥ two CTCs per 7.5 mL blood, *p* = 0.155) and NSCLC patients (Appendix A; cut-off ≥ one CTCs per 7.5 mL blood, *p* = 0.501).

### 2.2. Expression of CD74 and CD44 on CTCs

To assess a possible role of CD74 and CD44 expression in BM formation, expression of both proteins on CTCs was calculated as a total immuno-fluorescence signal intensity (mean grey value) per cell and fluorescence channel. Although almost all patients with CTCs had cells expressing both CD74 and CD44, a great heterogeneity between the individual cells was observed. In the breast cancer BM cohort, all seven CTC positive patients showed a mostly strong or intermediate expression for CD74 (39/40 CTCs, Figure 2B) as well as for CD44 (3/40 CTCs, Figure 2C). Only patient four (P4) had one CTC lacking both CD74 and CD44 expression. CD74 expression strongly correlated with CD44 expression on CTCs from BC-BM patients (Figure 2D, Pearson’s r = 0.9686). A moderate correlation between either CD74 or CD44 and total keratin expression could be observed (Pearson’s r = 0.6164 and r = 0.7653, respectively; data not shown).

Similar to breast cancer, the majority of CTCs enriched from NSCLC-BM patient’s blood showed a strong expression for both CD74 (13/27 CTCs) and CD44 (15/27 CTCs). Two patients (P1, P2) did not show any CD74 expression, and only weak expression of CD44 (P3, P5; Appendix A D,E). A positive association of both proteins was observed (Pearson’s r = 0.8871, Appendix A), whereas no association to keratin expression was observed (Pearson’s r = 0.0724 and r = 0.3067, respectively; data not shown). Similar observation were determined in four CTC-positive melanoma-BM patients; most CTCs showed a strong expression of CD74 (15/26) and CD44 (19/26), with only one patient not showing any expression of either protein (Appendix A). Again, correlation analysis revealed an association of both proteins (CD44 and CD74) (Appendix A; Pearson’s r = 0.7986) and none to PMEL17/Melan-A+ expression (Pearson’s r = 0.2110 and r = 0.1492, respectively; data not shown). The expression of either CD74 or CD44 (by subdividing the CTC-positive cases in marker positive and negative) did not add to the prognostic power of CTCs in none of the patient cohorts (data not shown).

### 2.3. Correlation of CD74 and CD44 Expression on CTCs and Matched BM Tissue

In eleven breast cancer patients, blood was taken just before BM surgery. In these cases, a BM tissue sample was available for immunohistochemical analyses of CD74 and CD44 protein expression. Expression profiles of both proteins were determined by using the same H-score as described in Zeiner et al. [14]. Positive CD74 expression was observed as membranous and cytoplasmic staining, whereas CD44 was only expressed at the cell membrane (Figure 2E).

In four out of the eleven matched samples, CTCs could be identified. Interestingly, in these cases, the CD74 and CD44 protein expression in the matched BM sections showed a different expression pattern compared to the CTCs (Figure 2F,G). Many of the BM samples were totally negative for either CD74 (54%) or CD44 (46%) protein expression whereas the matching CTCs were positive. Similar to the CTC results, in those BM samples in which the tumor expressed either marker, the expression showed a great heterogeneity between the cells (Figure 2F,G).

### 2.4. Detailed Analysis of Blood, CSF and BM Tissue of a HER2-Positive Breast Cancer Patient

Breast cancer patient P6, a HER2 and hormone receptor-positive breast cancer patient (HER2+, ER+, PR+), with additional metastases in bone, lung and liver, had 16 detectable CTCs, all with a strong or intermediate expression of both CD74 and CD44 (Figure 3A). The subsequent copy number variant (CNV) analysis of nine single CTCs demonstrated three groups of genetically slightly different profiles with differentiating aberrations at three chromosomal loci. The gain of complete chromosome 4 was seen in three CTCs. The loss of 14q was seen in five other CTCs. Two of these CTCs had no loss of 11q that was seen in all other CTCs. All CTCs had a high-level gain of the HER2 loci (chromosome 17p12-21.32) (Figure 3D and Appendix A). The NGS analyses identified mutations in eight cancer associated genes: *ESR1*, *MECOM*, *JAK3*, *KMT2C*, *CTNNB1*, *CALR*, *NCOR1* and *AMER* (Figure 3E). None of the mutations were seen in all of the CTCs. Interestingly, *ESR1* was found mutated in half of the cells, indicating a potential mechanism for the relapse of the patient. The identified mutations did not correlate with a certain CNV profile, with one exception: in 7/8 (88%) of analyzed CTCs, a mutation in *MECOM* could be identified and this mutation correlated with one CNV profile (Figure 3D). This patient had a leptomeningeal spread. Interestingly, both the solid tumor mass and the 11/12 identified disseminated tumor cells (DTCs and DTC cluster) in CSF were negative for both CD74 and CD44 expression (Figure 3B,C). Only one solitary DTC in the CSF showed CD74 expression, whereas it was negative for CD44, indicating that the CSF DTC sample seemed to resemble the solid tissue expression pattern in contrast to the CTCs found in blood circulation.

## 3. Discussion

CTC enrichment from BM patients is challenging as most CTCs shed by BM have been shown to be of a more mesenchymal phenotype, including negative for EpCAM [8,10,19]. Here we show that using a microfluidic marker-independent device, CTCs can be found in a large fraction of BM patients. A total of 50% of breast cancer and NSCLC samples showed CTC, whereas 36% of melanoma patients, had CTCs. The CTC positivity rates gained through this approach were clearly higher especially in NSCLC cases with brain metastases compared to other studies using, e.g., the EpCAM-dependent CELLSEARCH^®^ system, with CTC positivity rates of only 12.5% [21]. Similarly, also in breast cancer BM patients, CELLSEARCH CTC positivity rates of 20–22% have been reported [10,22,25]. As far as we know, no study has been published on CTCs specifically in BM melanoma patients. In general, due to the large heterogeneity in melanoma marker expression and, thus, the lack of a gold standard method for CTC detection, the published detection rates vary substantially ranging between 1 and 62% [20,23,24,27]. Obviously most breast and melanoma patients also had other distant metastases in addition to the brain. Therefore, in these cases, the origin of CTC is not clear. However, in the NSLC cohort, several patients with only brain metastases were included. In this oligo-metastatic cohort, the CTC rates and CD44 and CD74 marker expression did not differ from the multi metastatic cases, suggesting that both brain metastases and other metastases shed CTCs into the circulation and these CTCs can express CD74/44.

Here, the CTCs were detected using a novel semi-automatic CTC detection and quantification protocol [13]. Manual screening and enumeration of CTCs is time consuming and clearly subjective and, thus, may introduce bias into the detection process. Using the automated Xcyto^®^ Quantitative cell imager, which makes use of defining thresholds of the different expression intensities of different cell populations, bypasses this subjective influence. Here, CTCs were defined as positive for either pan-keratins for breast cancer and NSCLC or PMEL17/Melan-A-positive for melanoma samples, positive nucleus signal (DAPI) and negative for CD45 (leukocyte exclusion marker). In a few cases, non-typical keratin or PMEL17/Melan-A-positive and CD45-negative cells with either an abnormal larger size and/or multinucleated could be detected as well. Although keratin-positive/CD45-negative, these cells might be polyploid hematopoietic cells or megakaryocytes, the latter, however, having been shown to predict poor prognosis in NSCLC [26,28]. This study was too small to define their potential clinical relevance alone. Here, all patients with these large cells also had normal-sized “classical” CTCs and, therefore, no bias in the clinical association analyses was, thus, inferred.

Expression analysis of CD74 and CD44 showed that most CTCs show a strong or intermediate expression of these two markers with no influence on patient outcome. The expression of both proteins showed a highly correlative expression, supporting the previously published data on a direct interaction between CD74 and CD44 in breast cancer [12,29].

To the best of our knowledge, this study is the first describing CD74 expression on CTCs, whereas CD44 has been studied before. CD44 has been proven to be commonly expressed on CTCs and linked with both cluster as well as metastatic initiating capabilities [30,31]. CD44 was shown to be expressed on both single CTCs as well as in CTC clusters and this clustering was dependent on homophilic interactions of CD44. Furthermore, the study suggested that CD44-negative CTCs undergo anoikis within 48–72 h after detachment from the tumor [31]. Here, we could identify CD44 on the majority of CTCs in the BM patients; therefore, possibly indicating that most of the detected CTCs in BM patients are potential seeds for new metastasis. Future studies are needed to confirm the possible difference in CD74 and CD44 expression in patients with different metastatic patterns. Moreover, the expression of primary tumors and different metastases from the same patients would be important to assess.

Interestingly, in contrast to CTCs, most matched BM tissue samples were negative for CD74 and CD44 and, thus, no correlation between CD74 and CD44 tumor tissue and CTC expression was found. Obviously, the number of matched samples was very limited. Still, the more heterogeneous expression pattern of both proteins as well as the ratio of positive/negative tumor cells in breast cancer BM is in line with other studies [15,16,17]. In one patient with leptomeningeal spread, both the tissue and the CTCs and DTC in blood and CSF were analyzed. Interestingly, here the CD74 and CD44 expression on DTCs in CSF and tissue resembled each other being mainly negative, whereas the peripheral blood CTC profiles were totally different, showing a strong/intermediate expression of both proteins, again supporting the perhaps different role of these protein in the brain environment and in circulation. NGS analysis of nine individual CTCs identified three different CNV profiles and mutations in eight cancer associated genes, including a potentially resistance-associated ESR1 mutation.

The different expression patterns observed between the brain environment and peripheral blood could point to different biological roles of these two proteins in CTCs compared to BM tissue. As previously mentioned, CD44 has been shown to be crucial for CTC survival in the blood flow, whereas CD44 might not be required for BM growth; furthermore, downregulation of CD44 in BM tissue might be the result of chemo- and radiotherapy, respectively [31,32]. Similarly, CD74 might be upregulated in the blood flow, as there are numerous cytokines and leukocytes present which are able to induce CD74 expression [33]. In contrast, we have shown that in BM tissue loss of CD74, there is a strong negative prognostic marker with the expression regulated by promoter methylation and negatively associated with tumor-infiltrating T-lymphocytes (TILs) [14]. Thus, future studies are necessary to show whether this CD74 and CD44 induction on CTCs could be supportive of an active anti-tumor immune response in the blood flow.

In summary, we show that in BM patients a size based microfluidic approach in combination with a semi-automated cell recognition system is well suited for CTC detection, including clusters and more sensitive comparable to EpCAM-based enrichment systems. Furthermore, our study demonstrates that most patients with BM have CTCs that express both CD74 and CD44, while the CD74 and CD44 expression in corresponding BM tissue is less frequent and more heterogeneous. Therefore, our results suggest a potential role of CD74 and CD44 in survival and trafficking of CTCs in the circulation. However, this hypothesis needs further testing in a larger cohort of patients, including non-BM patients as well as a further detailed analysis of detected CTCs.

## 4. Material and Methods

### 4.1. Patient Materials

Whole blood samples of forty-four BM patients (14 breast cancer, 18 NSCLC and 11 melanoma) were collected prospectively and screened for CTCs using the microfluidic size-based CTC enrichment system Parsortix^®^ (ANGLE plc, Surrey, UK). Blood collection was performed either before clinical resection of the BM (breast cancer 10/14, NSCLC 17/18 and melanoma 5/11) or prior to the start of a new line treatment (breast cancer 4/14, NSCLC 2/18 and melanoma 6/11). Average time between primary tumor and BM diagnosis was 7.5 years (range, 2–26 years) for breast cancer and 3.4 years (range, 0.3–13 years) for melanoma patients. In total, 12/18 NSCLC patients, BM diagnosis was the initial diagnosis of primary lung cancer with the remaining 7/18 having an average time of 1.1 years (range, 0.1–1.8 years) between PT and BM diagnosis. In total, 36% (5/14) of breast cancer, 74% (14/18) of NSCLC and 9% (1/11) of melanoma patients, BM were the only metastatic site (oligo-BM). The median follow-up time after blood collection was 20.6 months (1.0–16.3 months) for breast cancer, 14.5 months (1.0–55.4 months) for NSCLC and 6.0 months (1.0 –16.3 months) for melanoma patients. For a total of 11/14 breast cancer patients, whole sections of formalin-fixed, paraffin-embedded (FFPE) matched brain metastatic tissue were available. All patients were treated at the Hamburg University Medical Center. Ethical approval for this study was granted by the IRB Ethical Review Board of Hamburg analyses of human materials (PV4954) and all participants gave their written consent for the study.

### 4.2. Immunohistochemistry of Brain Metastatic Samples

FFPE tissue slides of brain metastatic samples were baked for 2 h, deparaffinated with xylene and gradually hydrated with ethanol. Slides were boiled for 5 min in a pressure cooker in Antigen Retrieval Citra Plus Solution (HK080-9K, BioGenex, Fremont, CA, USA) for CD74 or Target Retrieval Solution Citrate (S1699, Dako-Agilent, Santa Clara, CA, USA) for CD44, respectively. After cooling, primary antibodies for CD74 (1:100, clone LN2, ab9154, Abcam, Cambridge, UK) and CD44 (1:100, clone G44-26, 560977, BD Bioscience, Heidelberg, Germany) were applied over night at 4 °C. Detection of primary antibodies was performed by the Dako REAL Detection System (Peroxidase/ DAB+, K5001, Dako-Agilent, Santa Clara, CA, USA) according to the manufacturer’s protocol and counterstained with hematoxylin. CD74 and CD44 positive tumor cells were analyzed using a semi-quantitative IHC H-score as described in Zeiner et al. [14]. In brief, staining intensity level (1 = weak, 2 = moderate, 3 = strong) and the percentage of positively stained tumor area per whole tissue samples were multiplied, resulting in a final range from 0 to 300 with an H-score < 10 considered as negative, 11 ≤ H-score < 100 as weak and an H-score ≥ 100 as strong expression. The results for CD74 and CD44 were independently scored by two experienced researchers (P.N.H. and D.L.) and, in case of discrepant finding, re-evaluated together.

### 4.3. Detection of Circulating Tumor Cells and Disseminated Tumor Cells (CTCs and DTCs)

An amount of 7.5 mL of peripheral blood were collected in EDTA tubes and processed immediately by Parsortix^®^ using the protocol as before [13], followed by a cytocentrifugation of the obtained cells onto slides. Additionally, one cerebrospinal fluid (CSF) sample (1mL) from a breast cancer patient, cytocentrifuged directly and without further processing on a slide, was used for DTC detection. For visualization, cells were fixed with 4% paraformaldehyde for 10 min at room temperature, followed by 10 min permeabilization with TX-100 and subsequent blocking with 10% AB-serum (#805135, Bio-Rad Laboratories Inc., Hercules, CA, USA). Primary antibody against CD74 (1:200, clone LN2, ab9154, Abcam, Cambridge, UK) was applied in 10% AB-serum o/n at 4 °C and conjugated with PerCP goat anti-mouse secondary antibody (1:500, F0114, R&D Systems Inc., Minneapolis, MN, USA) for 1h at room temperature. Subsequently, human keratins (1:100, clone AE1/AE3, 53-9003-82, eBioscience, San Diego, CA, USA) for breast cancer and NSCLC samples, PMEL17 (1:100, clone HMB45, NBP2-34638AF488, Novus Biologicals, Littleton, CO, USA) and Melan-A/Mart-1 (1:100, clone CM2-7C10, NBP2-33148AF488, Novus Biologicals, Littleton, CO, USA) for melanoma samples, as well as CD44 (1:100, clone G44-26, 561858, BD Biosciences, Heidelberg, Germany) and CD45 (1:150, clone HI30, 304012, BioLegend, San Diego, CA, USA) were applied in 10% AB-serum for 1h at room temperature. Nuclei were stained with DAPI (1 µg/mL, Sigma-Aldrich, St. Louis, MO, USA). CTCs were defined as keratin+ (breast cancer and NSCLC) or PMEL17/Melan-A+ (melanoma)/DAPI+/CD45- using the Xcyto^®^ Quantitative cell imager (ChemoMetec, Allerod, Denmark) as described in Koch et al. [13]. In brief, cytospins were scanned by a 4× magnification. Cells were identified due to the positive DAPI signal of the nuclei and CTCs by plotting the intensities of AF488 (tumor cell markers) against AF647 (CD45). Gating for CTCs was achieved for cells with a high AF488 signal and low AF647. As a result, a gallery of possible CTCs was established and a picture 20× of each as CTC-evaluated cell was performed to identify real positive hits. Total protein expression for CD74 and CD44 was measured as mean grey values of fluorescence intensity in either channel by using the software Fiji. Expression was categorized as negative (mean grey value ≤ 10), weak (11 ≤ mean grey value > 20) and strong (mean grey value ≥ 21).

### 4.4. Whole Genome Amplification (WGA) and Quality Control of CTCs

Single CTCs were picked by micromanipulation (micro injector CellTram Vario and micromanipulator TransferManNKII, Eppendorf Instruments, Hamburg, Germany). The genomes of the picked cells were amplified by whole genome amplification (WGA) using the Ampli1TM WGA Kit for single cells (Menarini Silicon Biosystems, Florence, Italy) and the quality of the WGA product was assessed by multiplex PCR of the *GAPDH* gene producing 96, 108–166, 299 and 614 bp fragments using the Ampli1™ QC Kit (Menarini Silicon Biosystems) as described before [13]. The PCR products were analyzed using a 2% agarose TAE gel, and samples producing three or four bands were chosen for the NGS analyses. As a positive control, human leukocyte DNA was used.

### 4.5. Whole Genome Sequencing

For library preparation and further analysis, single cells from P6 were used only as the others did not reach required amplification quality. For low coverage, WGS libraries were prepared using 50 ng and the Twist Human Core Exome EF Multiplex Complete Kit (#100803, Twist Bioscience, San Francisco, CA, USA), but omitting the exome enrichment step. Paired-end libraries were sequenced on HiSeq 4000 and NovaSeq 6000 instruments (2 × 75 bp or 2 × 100 bp, respectively) to ×1 coverage.

### 4.6. Data Processing

Raw reads were subjected to adapter and quality trimming with the BBDuk tool which is a part of the BBTools package (BBMap—Bushnell B.—sourceforge.net/projects/bbmap/) (version 37.90; parameters: minlen = 25, qtrim = rl, trimq = 10, ktrim = r, k = 25 mink = 11, hdist = 1, overwrite = t; Nextera adapters clipped from both reads). The processed reads were aligned to the human genome (hg19) using BWA in “mem” mode [34] (version 0.7.17-r1188; parameters: -L 0 -M) keeping only the primary alignments. Duplicates were removed using the MarkDuplicates tool a part of Picard Toolkit (Broad Institute, GitHub Repository; http://broadinstitute.github.io/picard/ (accessed on 14 October 2020)) (version 2.17.11). Raw sequencing quality was assessed with the FastQC 0.11.8. (Available online: https://www.bioinformatics.babraham.ac.uk/projects/fastqc/ (accessed on 14 October 2020)) whereas the FastQ Screen 0.13.0 (Available online: https://www.bioinformatics.babraham.ac.uk/projects/fastq_screen/ (accessed on 14 October 2020)) tool was used to control possible contamination.

The ichorCNA 0.3.2 (Available online: https://github.com/broadinstitute/ichorCNA (accessed on 14 October 2020)) was used to quantify tumor fraction in DNA from lcWGS without prior knowledge of somatic single nucleotide variants (SSNVs) or SCNAs present in the primary tumor sample. Further details are provided within the Appendix A. Establishment of the ichorCNA tool was originally described by Adalsteinsson [35].

### 4.7. Mutation Analysis

Mutation analysis was performed using Strelka2 (2.8.4; [36]) per sample against the reference genome. Mutations were further annotated using the Ensembl Variant Effect Predictor (v99; [37]). Mutations with population frequencies from 1000 genomes phase 3 and gnomAD v2.1 larger than 0.001 were classified as germline. A background list of mutation calls from 40 other unpublished CTC samples was used to identify and label recurrent mutation positions as potential artifacts.

### 4.8. Statistics

Data are presented as a percentage or median ± interquartile range. Statistical analysis of patient samples was performed using SPSS 23.0 and in silico Online v2.2.1 [38]. The correlation of clinical and pathological variables with the staining was examined using Barnard’s unconditional multinomial exact test with Boschloo’s statistic. Kaplan-Meier survival curves were compared with the log-rank test. Statistical analysis of CTC correlation analysis was performed using the GraphPad Prism 8.0. For simple comparison, the Mann–Whitney test was performed. A *p* < 0.05 was defined as significant.

## Figures and Tables

**Figure 1 ijms-22-06993-f001:**
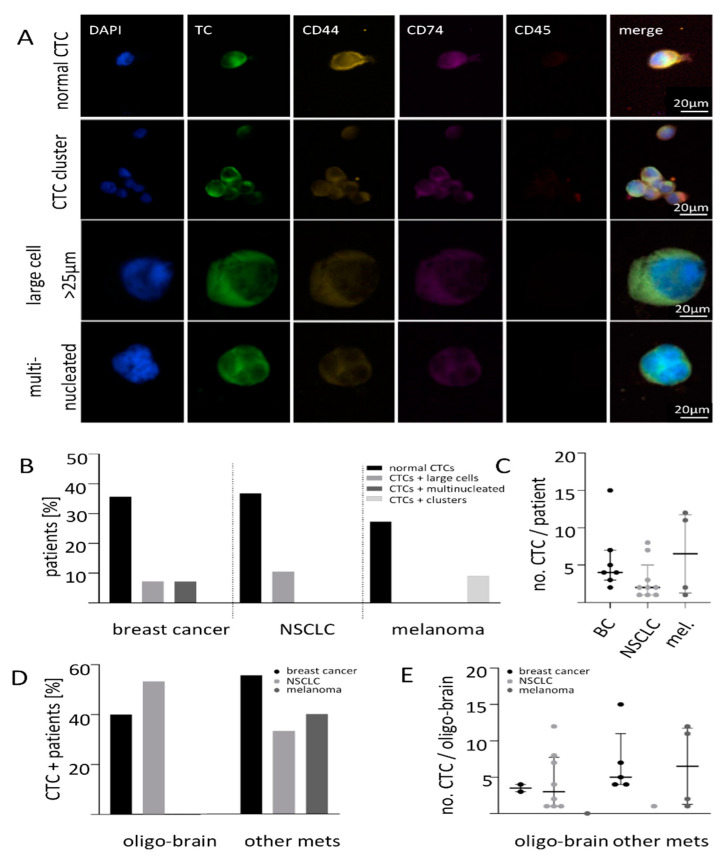
Liquid biopsy analysis of brain metastatic patients. (**A**) Xcyto^®^10 analysis of liquid biopsies from patients with breast cancer, NSCLC and melanoma revealed different kind of CTCs enriched with Parsortix from 7.5 mL whole blood. (**B**) CTCs were detected in 50% of breast cancer, in 47% of NSCLC and in 36% of melanoma patients with mostly typical CTCs (black). The CTC positive patients were further subdivided as only normal CTCs, normal CTCs and large cells, normal CTCs and multinucleated cells as well as normal CTCs and clusters. Additionally, CTC clusters could be detected in the melanoma cohort (light grey) and some patients had tumor marker-positive large cells (mid grey) or multinucleated cells (dark grey). (**C**) No difference in the average number of CTCs between the different tumor entities. (**D**) Patient cohort divided by the metastatic spread shows varying CTC positivity. (**E**) No difference in the median CTC numbers could be observed in relation to the metastatic site (breast cancer *p* = 0.160; NSCLC, *p* = 0.419, melanoma *p* = 0.400).

**Figure 2 ijms-22-06993-f002:**
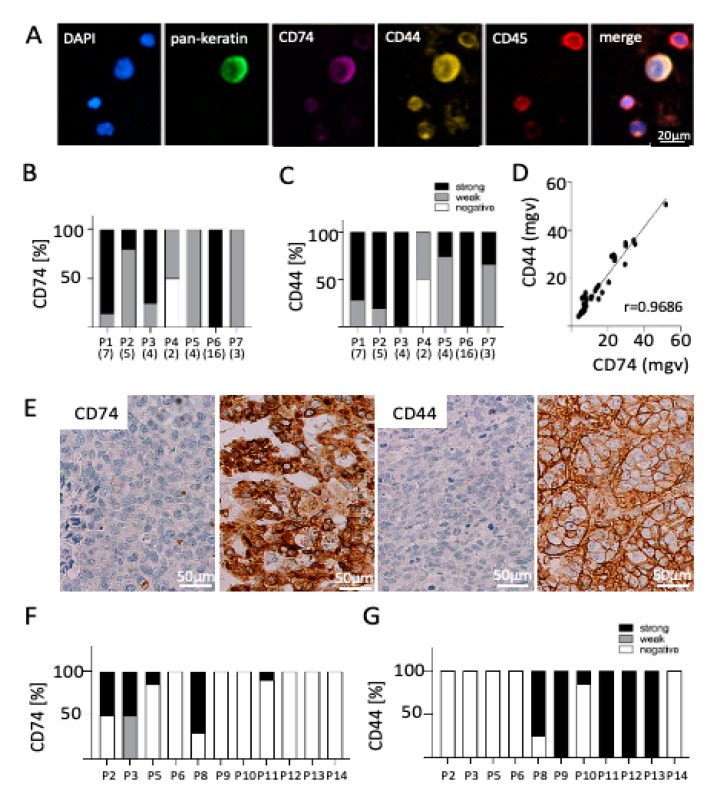
CD74 and CD44 expression on CTCs and matched BM in breast cancer. (**A**) Exemplary CTC positive for pan-keratin, CD74, and CD44 and negative for CD45 (leukocyte exclusion marker) surrounded by four leucocytes positive for CD45 and for both CD44 and CD74. Bar chart showing the (**B**) CD74 and (**C**) CD44 expression on CTCs in seven breast cancer BM patients. Number in parenthesis represents the number of CTCs identified in each patient. Most patients showed a weak or high expression of both proteins on their CTCs. (**D**) Correlation analysis revealed that CD74 and CD44 are expressed to a similar extent on enriched CTCs (r = 0.9686; mgv = mean grey value). (**E**) IHC analysis of matched brain metastasis revealed differential expression of (**F**) CD74 and (**G**) CD44 independent of CTC status (P2–P6: CTC-positive patients, P8-P14 CTC-negative patients), determined by H-score (see method section for details). Each bar indicates the expression pattern of one brain metastasis sample per patient.

**Figure 3 ijms-22-06993-f003:**
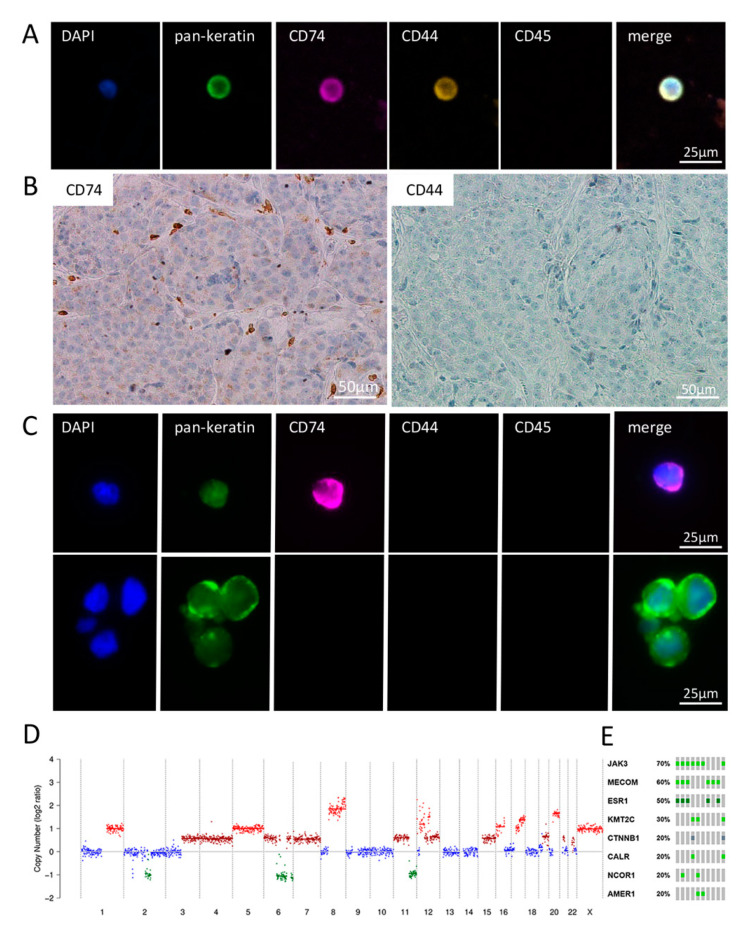
Matched samples of blood CTCs, BM and CSF-CTCs of breast cancer patient P6. (**A**) Exemplary CTC positive for keratins, CD74 as well as CD44 and negative for CD45 (leukocyte exclusion marker). (**B**) Matched BM tissue shows no expression of CD74 and CD44. (**C**) Matched liquor samples show single CTCs as well as clusters. Only one liquor DTC was positive for CD74, whereas all DTCs were negative for CD44 and CD45. (**D**) Copy number alteration profile of a single CTC of P6 showing numerous large chromosomal losses and gains, including a high-level amplification of chromosomes 8 (MYC loci) and 12 and 17 (HER2) seen in all CTCs. (**E**) Most common identified mutation in 10 CTCs analyzed (green: missense mutation (unknown significance); dark green: missense mutation (putative driver); grey: truncating mutation (unknown significance).

**Table 1 ijms-22-06993-t001:** Correlation of CTC status to clinical parameters of breast cancer, NSCLC and melanoma patients.

Breast Cancer			CTC				Lung Cancer		CTC			Melanoma			CTC		
			Negative	Positive					Negative	Positive					Negative	Positive	
	Total n	**14**	7	7			Total n	**18**	9	9			Total n	**11**	7	4	
			**%**	**%**	***p***				**%**	**%**	***p***				**%**	**%**	***p***
**Survival**					**0.155**	**Survival**					**0.501**	**Survival**					**0.049**
Alive		5	20.0	80.0		Alive		4	50.0	50.0		Alive		6	100.0	0.0	
Dead		9	66.7	33.3		Dead		14	57.1	42.9		Dead		5	20.0	80.0	
**ER**		**8**	37.5	62.5	**0.486**	**BRAF**		**0**	-	-	-	**BRAF**		**8**	50.0	50.0	**-**
**PR**		**6**	33.3	66.7	**0.486**	**EGFR**		**1**	0.0	100.0	**-**	**PD-L1**		**3**	66.7	33.3	**-**
**HER2**		**10**	40.0	60.0	**0.409**							**NRAS**		**3**	66.7	33.3	**-**
**BRCA1**		**2**	100.0	0.0	**-**												
**Metastases**						**Metastases**						**Metastases**					
Bone		**10**			**0.999**	Bone		**18**			**0.999**	Bone		6			**0.206**
	No	5	60.0	40.0			No	17	55.6	44.4			No	2	0.0	100.0	
	Yes	5	40.0	60.0			Yes	1	0.0	100.0			Yes	4	75.0	25.0	
Liver		**9**			**0.999**	Liver		**18**			**0.999**	Liver		8			**0.003**
	No	7	42.9	57.1			No	17	55.6	44.4			No	3	0.0	100.0	
	Yes	2	50.0	50.0			Yes	1	100.0	0.0			Yes	5	100.0	0.0	
Lung		**10**			**0.999**	Lung		**18**			0.999	Lung		8			**-**
	No	5	40.0	60.0			No	17	55.6	44.4			No	0	-	-	
	Yes	5	40.0	60.0			Yes	1	100.0	0.0			Yes	8	50.0	50.0	
Lymph node		**11**			**0.437**	Lymph node		**18**			0.999	Lymph node		8			**-**
	No	7	42.9	57.1			No	10	54.5	45.5			No	0			
	Yes	4	75.0	25.0			Yes	8	62.5	37.5			Yes	8	62.5	37.5	
Other organ		**9**			**0.487**	Other organ		**18**			0.999	Other organ		9			**0.999**
	No	5	40.0	60.0			No	15	50.0	50.0			No	1	100.0	0.0	
	Yes	4	75.0	25.0			Yes	3	100.0	0.0			Yes	8	50.0	50.0	
Oligo brain		**9**			**0.999**	Oligo brain		**18**			0.394	Oligo brain		11			**0.999**
	No	7	42.9	57.1			No	3	54.5	45.5			No	10	60.0	40.0	
	Yes	2	60.0	40.0			Yes	15	47.0	53.0			Yes	1	100.0	0.0	

## Data Availability

All datasets are presented in the main manuscript and Appendix A.

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
