# Peer review of "CD74 and CD44 Expression on CTCs in Cancer Patients with Brain Metastasis"

_ijms, 2021, doi:10.3390/ijms22136993_

Round 1
Reviewer 1 Report
Brain metastasis (BM) has a significant impact on mortality in cancer patients and its incidence is increasing in many cancers. In this study, the authors isolated and characterized the circulating tumor cells (CTCs) in BM patients. They developed an efficient protocol of CTC isolation in BM patients and found that most patients have CTCs that express both CD74 and CD44. Interestingly, the expression of CD74 and CD44 is less frequent in BM tissue. These results raised an attractive hypothesis that CD74 and CD44 enhanced the survival and trafficking of CTCs in circulation.
I think that this article is suitable for publication in IJMS after minor revision.
Page16Line6; tissues. As >>> tissues. As
Figure1B/D/E, Figure2C/G, Supplementary Figure 2E/H; Legends of these charts are all unclear and should be revised.
Figure legends for Fig.1D/E, Fig.2E/F/G, and Fig.3E should be added.
Figure legend of Fig.2; ex-pression >>> expression, pa-tients >>> patients, the on CTCs >>> the CTCs
Figure 3D/E; The resolution of these figures is insufficient and should be replaced.
Table1; Position of “14” of breast cancer’s “total n” is incorrect.
Author Response
Dear Reviewer 1
Thank you very much for the positive review and giving us a chance for revisions.
We have now answered in details all issue raised by the reviewers. Please find them highlighted in bold below your comments / suggestions. All changes done in the manuscript are highlighted in yellow. As already communicated directly after submission, we sadly made an error during the submission and omitted two authors that both contributed seminally for the production of this manuscript. We have also sent the manuscript for language revision as it was suggested by a reviewer.
We hope and believe we were able to answer all issue raised be the 4 reviewers and we sincerely hope that our manuscript now full fills the criteria of International Journal of Molecular Sciences.
Reviewer 1:
Open Review
Brain metastasis (BM) has a significant impact on mortality in cancer patients and its incidence is increasing in many cancers. In this study, the authors isolated and characterized the circulating tumor cells (CTCs) in BM patients. They developed an efficient protocol of CTC isolation in BM patients and found that most patients have CTCs that express both CD74 and CD44. Interestingly, the expression of CD74 and CD44 is less frequent in BM tissue. These results raised an attractive hypothesis that CD74 and CD44 enhanced the survival and trafficking of CTCs in circulation.
I think that this article is suitable for publication in IJMS after minor revision.
Thank you very much for your comments. We have now addressed all points raised by you.
Page16Line6; tissues. As >>> tissues. As
We have changed accordingly.
Figure1B/D/E, Figure2C/G, Supplementary Figure 2E/H; Legends of these charts are all unclear and should be revised.
The quality of all figures has been improved and therefore also the legend are clear now.
Figure legends for Fig.1D/E, Fig.2E/F/G, and Fig.3E should be added.
We are extremely sorry for the incomplete figure legends. Unfortunately, the legends were accidentally shortened when the Journal was converted the manuscript into their template. We already informed the Journal about this. All figure legends are now complete.
Figure legend of Fig.2; ex-pression >>> expression, pa-tients >>> patients, the on CTCs >>> the CTCs
We have changed accordingly.
Figure 3D/E; The resolution of these figures is insufficient and should be replaced.
Please see comment above. The resolution for all figures has been improved.
Table1; Position of “14” of breast cancer’s “total n” is incorrect.
Thank you very much for this note. We corrected the position.
Reviewer 2 Report
Loreth et al present an exploratory study on CD74 and CD44 expression on CTCs of patients with brain metastatic cancer (NSCLS, melanoma and breast cancer) using a size exclusion based microfluidic platform in combination with semi-automated cell detection. As their key findings, authors show that the expression of CD44 and CD47 correlates to each other and that most of the detected CTCs were positive for both markers, whereas matching brain metastatic tissue was mostly CD44/CD74 negative. The study is well written and interesting. However, in order to publish this work in the International Journal of Molecular Science, the following concerns need to be addressed by the authors:
Major:
1.) The link, the authors suggest between CD74/CD44-positive CTCs and brain metastases are not clearly supported by the presented data and could be considerably confounded by the fact that blood samples were taken both i) at the time of initial diagnosis of brain metastasis and ii) at the time of brain metastasis after long (tumor free?) latency periods (2-26 years). Moreover, it is stated in the methods section that in only 36% of the breast cancer patients, in only 74% of the NSCLC patients and in only 9% of the included melanoma patients, brain metastases were the only metastatic site. Therefore, the analysed patient cohort is principally not fully compatible with the study objective, as introduced in the abstract: “Here, we assessed whether circulating tumor cells (CTCs) in the peripheral blood can serve as disease surrogate, focusing on CD44 and CD74 expression as markers for BM prognosis.” Please adapt study objective, conclusions and argumentation throughout the manuscript.
2.) The hypothesis that CD44/C74 expression on CTCs may be associated with CTC plasticity and a possible “in transit state” in the blood needs to be confirmed by additional immunohistochemical analysis of the CD44/CD47 expression in the primary tumor and in other available metastatic sites (e.g. visceral metastases, bone metastases, whatever was found in those patients, who had also other metastases). It could be also possible, that CD74/CD44 positive CTCs simply do not have brain tropism and have their origin from other (residual, dormant, or active) metastatic sites.
Minor:
1.) Introduction line 21, please remove “and thus”
2.) Figure quality is poor, please increase resolution.
Author Response
Dear Reviewer 2,
Thank you very much for the positive review and giving us a chance for revisions.
We have now answered in details all issue raised by the reviewers. Please find them highlighted in bold below your comments / suggestions. All changes done in the manuscript are highlighted in yellow. We have also sent the manuscript for language revision as it was suggested by a reviewer.
We hope and believe we were able to answer all issue raised be the 4 reviewers and we sincerely hope that our manuscript now full fills the criteria of International Journal of Molecular Sciences.
Loreth et al present an exploratory study on CD74 and CD44 expression on CTCs of patients with brain metastatic cancer (NSCLS, melanoma and breast cancer) using a size exclusion based microfluidic platform in combination with semi-automated cell detection. As their key findings, authors show that the expression of CD44 and CD47 correlates to each other and that most of the detected CTCs were positive for both markers, whereas matching brain metastatic tissue was mostly CD44/CD74 negative. The study is well written and interesting. However, in order to publish this work in the International Journal of Molecular Science, the following concerns need to be addressed by the authors:
Major:
-
- The link, the authors suggest between CD74/CD44-positive CTCs and brain metastases are not clearly supported by the presented data and could be considerably confounded by the fact that blood samples were taken both i) at the time of initial diagnosis of brain metastasis and ii) at the time of brain metastasis after long (tumor free?) latency periods (2-26 years). Moreover, it is stated in the methods section that in only 36% of the breast cancer patients, in only 74% of the NSCLC patients and in only 9% of the included melanoma patients, brain metastases were the only metastatic site. Therefore, the analysed patient cohort is principally not fully compatible with the study objective, as introduced in the abstract: “Here, we assessed whether circulating tumor cells (CTCs) in the peripheral blood can serve as disease surrogate, focusing on CD44 and CD74 expression as markers for BM prognosis.” Please adapt study objective, conclusions and argumentation throughout the manuscript.
Reply: We thank the reviewer for this comment. We agree with the reviewer that in patients with multiple metastases one cannot differentiate the specific metastatic source of CTCs. Our cohort is rather small but representative: oligo brain met disease are commonly found in NSCLC, while being extremely rarer in breast and melanoma patients.
Interestingly, despite the blood brain barrier, we observed no differences in either the total CTC counts or the CD44 or CD74 expression between oligo and poly-metastatic NSCLC patients, which suggests that both brain metastases and other metastases shed CTCs into the circulation and these CTCs can express CD74/44. We have now discussed this issue in more detail in the manuscript.
- The hypothesis that CD44/C74 expression on CTCs may be associated with CTC plasticity and a possible “in transit state” in the blood needs to be confirmed by additional immunohistochemical analysis of the CD44/CD47 expression in the primary tumor and in other available metastatic sites (e.g. visceral metastases, bone metastases, whatever was found in those patients, who had also other metastases). It could be also possible, that CD74/CD44 positive CTCs simply do not have brain tropism and have their origin from other (residual, dormant, or active) metastatic sites.
Reply: Please see also our response to Comment 1 above. We agree with the alternative interpretation of our data by the reviewer and have added it to our revised Discussion. However, we like to add that other non-brain metastases are much less frequently operated than brain metastases in our and other Medical Centers. Thus, tissue from non-brain metastases was very hard to obtain for us.
Minor:
- Introduction line 21, please remove “and thus”
Thank you very much for this note. We removed the second “and thus” from the sentence.
- Figure quality is poor, please increase resolution.
The quality of all figures has been improved.
Reviewer 3 Report
Loreth and coauthors showed in their study that in brain metastasis cancer patients, most CTCs express CD74 and CD44, suggesting that CD74+/CD44+ CTCs were relevant to BM. Obtained results were clinically significant and may impact the relevant therapeutic regimen in the future. It will be helpful if authors could show how CD74 and CD44 were expressed on CTCs in non-BM cancer patients. Statistically significant difference of CD74+/CD44+ CTCs between the BM and non-BM cohorts will ensure robustness of authors' conclusion. This point should be addressed in Discussion.
Author Response
Dear Reviewer 3,
Thank you very much for the positive review and giving us a chance for revisions.
We have now answered in details all issue raised by the reviewers. Please find them highlighted in bold below your comments / suggestions. All changes done in the manuscript are highlighted in yellow. As already communicated directly after submission, we sadly made an error during the submission and omitted two authors that both contributed seminally for the production of this manuscript. We have also sent the manuscript for language revision as it was suggested by a reviewer.
We hope and believe we were able to answer all issue raised be the 4 reviewers and we sincerely hope that our manuscript now full fills the criteria of International Journal of Molecular Sciences.
Reviewer 3:
Loreth and coauthors showed in their study that in brain metastasis cancer patients, most CTCs express CD74 and CD44, suggesting that CD74+/CD44+ CTCs were relevant to BM. Obtained results were clinically significant and may impact the relevant therapeutic regimen in the future. It will be helpful if authors could show how CD74 and CD44 were expressed on CTCs in non-BM cancer patients. Statistically significant difference of CD74+/CD44+ CTCs between the BM and non-BM cohorts will ensure robustness of authors' conclusion. This point should be addressed in Discussion.
We thank the reviewer for this good comment. We agree with the reviewer that we would have needed to asses a second cohort of patients that did not have brain metastases in order to show the brain specific effects. We have now discussed this issue in more detail the Discussion.
Reviewer 4 Report
In the present manuscript the authors provide results from a study on CD44 and CD74 protein expression in CTCs, which were enriched from blood samples from breast cancer, NSCLC, and melanoma patients with brain metastases. In 11/14 breast cancer patients, a tissue sample from the brain met was available for IHC, and in a single case, cerebrospinal fluid was available for IF on tumor cells in this specimen. All blood samples were enriched using the label-free Parsortix technology. Keratin-positive cells were found in each 50% of the BC and NSCLC patients, and PMEL17/Melan-A/Mart-1-pos. cells in36% of the melanoma patients. In BC, these cells were prognostic. CD44 and CD74 protein expression was found in the majority of the CTC-pos. pat. CTCs were prognostic only in BC, and neither CD44 nor CD74 added prognositc value. In the tissue of brain metastasis from breast cancer pat, the CD44 and CD74 protein expression was more heterogenous than in the CTCs. In a single BC patient, NGS of 9 CTCs revealed CNV in 3 loci, and mutations in 8 genes.
Major comments:
- control group for CD74 and CD44 staining (healthy leukocytes)?
- Generally the quality of all figures needs to be improved, and the legends must be checked for completeness
- Figure 1:
- legend blurry
- legend is missing for 1D and 1E
- check aspect ratio
- Fig 1E: as in the text CTC in oligo-brain is compaired with CTCs in other mets, I would group these together for each tumor type.
- Results 2.1.: p-value in the text and in Fig 1E for melanoma subgroups is missing
- Results 2.2.
- from Fig 2B one cannot get the impression that all pat have "mostly strong expression for CD74". Indeed this is true for just 3/7 pat.
- a correaltion of r=0.6164 and 0.7653 is usually not considered as low but rather as moderate.
- are the CD44/CD74-neg CTCs included for the correlation analysis of these proteins with keratins or melanoma-specific proteins?
- add p-value for Pearson correlation
- how were the patients stratified for Kaplan-Meier analysis to assess the impact of CD44/CD74 on survival?
- Figure 2:
- check aspect ratios
- E-G legend is missing, and for D incomplete
- blurry legend in 2G
- Results 2.3.:
- as the Fig2 E-G legend is missing it is difficult to understand the results and verify the content of this abstract. From the present version it is not clear, if the 11 patients/blood samples are the same as in fig 2B? If yes, there is an overlap of just 4 pat with paired blood samples (Fig 1B,C) and BM (Fig 1F;G). Then, it is not admissible, to write "...showed a different expression patter..." and "...the expression showed a great heterogeneity...". For the comparison you should only consider paired samples (4 pat?)
- "The heterogenous expression pattern of both proteins..." belongs to the disussion.
- Results 2.4:
- is P6 identical to P6 from fig 2? If yes: she has 15 CTCs in Fig 2 and strong expr in all CTCs, here you mention 16 CTCs with strong or intermediate expression.
- "...the 11/12 identified CTCs..." does this number refer to the tumor cells detected in CSF? Aren't these cells commonly referred to as "DTC"?
- CNVs of all 9 CTCs would be interesting to have in Fig 3
- not every reader might know on which chromosome HER2 is located
- "None of the genes were seen..." rather "none of the mutations were seen..."?
- Figure 3:
- check aspect ratios
- E legend is missing, and for D incomplete
- improve quality of 3D and 3E (illegible)
- D shows CNV for a single CTC?
- Material an methods 4.3:
- how much volume of CSF was applied to the slide?
- was the sample processed using Parsortix?
- Suppl. fig 3: Legend does not fit to the image shown
Minor comments:
- introduction: review sentence " It is also expressed on many tumors..."
- section 4.6: the reference to Li et al is not included in the reference list. convert to citation
- Figure S2
- A-C harmonize layout
- increase quality (blurry)
- shift to the main text?
- Table 1:
- adding type of tumor to the top row of the table would increase the clarity
- add total number of CTC+ and CTC- pat in the top row for each tumor type
- total n for breast cancer is in the wrong line.
- for the genes, add the percentage of CTC+ cases for the wt
- for the NSCLC group, mostly the absolute numbers and not % are given.
- in the NSCLC group/survival check the % of the dead patients (40% from 14 is not an even number)
- Figure 2 legend: remove some "-" within the word; revise "(B) CD74 and (C) CD44..."
- Suppl. Fig 2D and E: total number of CTCs is 27, in the text it is 28
- Fig 3 legend: (leukocyte exclusion marker: ")" is missing. "(D) Copy number alteration profiles of single CTCs of P6 showing (a - omit) losses..."
Author Response
Dear Reviewer 4,
Thank you very much for the positive review and giving us a chance for revisions.
We have now answered in details all issue raised by the reviewers. Please find them highlighted in bold below your comments / suggestions. All changes done in the manuscript are highlighted in yellow. As already communicated directly after submission, we sadly made an error during the submission and omitted two authors that both contributed seminally for the production of this manuscript. We have also sent the manuscript for language revision as it was suggested by a reviewer.
We hope and believe we were able to answer all issue raised be the 4 reviewers and we sincerely hope that our manuscript now full fills the criteria of International Journal of Molecular Sciences.
In the present manuscript the authors provide results from a study on CD44 and CD74 protein expression in CTCs, which were enriched from blood samples from breast cancer, NSCLC, and melanoma patients with brain metastases. In 11/14 breast cancer patients, a tissue sample from the brain met was available for IHC, and in a single case, cerebrospinal fluid was available for IF on tumor cells in this specimen. All blood samples were enriched using the label-free Parsortix technology. Keratin-positive cells were found in each 50% of the BC and NSCLC patients, and PMEL17/Melan-A/Mart-1-pos. cells in36% of the melanoma patients. In BC, these cells were prognostic. CD44 and CD74 protein expression was found in the majority of the CTC-pos. pat. CTCs were prognostic only in BC, and neither CD44 nor CD74 added prognositc value. In the tissue of brain metastasis from breast cancer pat, the CD44 and CD74 protein expression was more heterogenous than in the CTCs. In a single BC patient, NGS of 9 CTCs revealed CNV in 3 loci, and mutations in 8 genes.
Major comments:
- control group for CD74 and CD44 staining (healthy leukocytes)?
Reply. As described by others and seen in Figure 2A, both CD74 and CD44 are not tumor specific markers but also expressed at various levels in different blood cells. Thus, in this manuscript we did not use these 2 markers for detecting CTCs. CTC were detected based on keratin (positive), dapi (positive) and CD45 (negative) expression as described in materials and methods, CD74 and 44 were only asses on CTCs. The staining of CD74 and CD44 were optimized using cell lines with known protein expression. We have now included a statement in figure 2A showing 4 leucocytes.
- Generally the quality of all figures needs to be improved and the legends must be checked for completeness
The quality of all figures has been improved and therefore also the legend are clear now. We are extremely sorry for the incomplete figure legends. Unfortunately, the legends were accidentally shortened when the Journal was converted the manuscript into their template. We already informed the Journal about this. All figure legends are now complete.
- Figure 1:
- legend blurry
please see comment above
- legend is missing for 1D and 1E
please see comment above
- check aspect ratio
please see comment above
- Fig 1E: as in the text CTC in oligo-brain is compaired with CTCs in other mets, I would group these together for each tumor type.-
- We are not sure what the reviewer means. We do have separated them according to tumor entity. Perhaps due to the extremely bad quality of images the headings could not be seen.
- Results 2.1.: p-value in the text and in Fig 1E for melanoma subgroups is missing
Thank you very much for this note. We added the p-value in the text as well as in Figure 1E.
- Results 2.2.
- from Fig 2B one cannot get the impression that all pat have "mostly strong expression for CD74". Indeed this is true for just 3/7 pat.
We have now changed the wording to “mostly strong or intermediate expression”.
- a correaltion of r=0.6164 and 0.7653 is usually not considered as low but rather as moderate.
Thank you very much for your comment. We have changed “low” to “moderate”.
- are the CD44/CD74-neg CTCs included for the correlation analysis of these proteins with keratins or melanoma-specific proteins?
The CD44/CD74 negative CTCs were included in the correlation analysis. As it is written in chapter 4.3, we categorized the expression of the proteins according to the following cut-off values: as negative (mean grey value £10), weak (11£ mean grey value > 20) and strong (mean grey value ³ 21).Therefore, also the CD44/CD74 “negative” values were included.
- add p-value for Pearson correlation
Thank you very much for your comment. We added the p-value also to the figure legend.
- how were the patients stratified for Kaplan-Meier analysis to assess the impact of CD44/CD74 on survival
In this analyses the CTC positive patients, were further sub divided into CD74/ CD44 negative and positive. We have clarified this in 2.2.
- Figure 2:
- check aspect ratios - please see comment above
- E-G legend is missing, and for D incomplete - please see comment above
- blurry legend in 2G - please see comment above
- Results 2.3.:
- as the Fig2 E-G legend is missing it is difficult to understand the results and verify the content of this abstract. From the present version it is not clear, if the 11 patients/blood samples are the same as in fig 2B? If yes, there is an overlap of just 4 pat with paired blood samples (Fig 1B,C) and BM (Fig 1F;G). Then, it is not admissible, to write "...showed a different expression patter..." and "...the expression showed a great heterogeneity...". For the comparison you should only consider paired samples (4 pat?)
We are very sorry that during the editing parts of the figure legend text was lost. Indeed here we show only data for the matched samples. We analyzed in total 11 samples with available data for blood and brain metastases. However, among these 11 matched pairs, only 4 patients had CTCs as shown in figure 2B and C. We admit that the correlation is thus based on a very small number of samples. This issue is now explained clearer in the text and discussed as a drawback in the discussion.
- "The heterogenous expression pattern of both proteins..." belongs to the disussion.
We have now moved this sentence to discussion.
- Results 2.4:
- is P6 identical to P6 from fig 2? If yes: she has 15 CTCs in Fig 2 and strong expr in all CTCs, here you mention 16 CTCs with strong or intermediate expression.
Thank you very much for your comment. We included “Breast cancer patient P6” to make clear that it is the same patient. We apologize for the typing error and corrected the number in the figure.
- "...the 11/12 identified CTCs..." does this number refer to the tumor cells detected in CSF? Aren't these cells commonly referred to as "DTC"?
Indeed for cells found in CSF both DTC as well as CTC is used In order to separate our peripheral blood CTC finding from the CSF we have now changed the CSF cells to DTCs.
- CNVs of all 9 CTCs would be interesting to have in Fig 3
We have now included all remaining CTCs in figure S3.
- not every reader might know on which chromosome HER2 is located
Thank you very much for this note. We add this information to the text (chromosome 17 p12 -21.32).
- None of the genes were seen..." rather "none of the mutations were seen..."? Thank you very much for this note. We changed to “mutations”.
- Figure 3:
- check aspect ratios - please see comment above
- E legend is missing, and for D incomplete - please see comment above
- improve quality of 3D and 3E (illegible) - please see comment above
- D shows CNV for a single CTC? – Yes figure 3D shows a CNV plot form one CTCs.
- Material an methods 4.3:
- how much volume of CSF was applied to the slide?
We received 1ml of CSF. This info is now included in materials and methods.
- was the sample processed using Parsortix?
To make this point more clear, we changed the sentence to „one cerebrospinal fluid (CSF) sample from a breast cancer patient was cytocentrifuged directly and without further processing on a slide was used for CTC detection”.
- fig 3: Legend does not fit to the image shown
We are sorry for this mistake. We have now added all remaining CNV plots and changed the heading: “Supplementary Figure S3: CNV plots of 8 single CTCs of breast cancer P6.”
Minor comments
- introduction: review sentence " It is also expressed on many tumors..."
We have now added the missing word “and”
- section 4.6: the reference to Li et al is not included in the reference list. convert to citation
We apologize for this. The reference is now added to the reference list.
- Figure S2
- A-C harmonize layout
We apologize for the disharmony. We correct this.
- increase quality (blurry) - please see comment above
- shift to the main text?
Thank you very much for this suggestions. As we can provide more information about breast cancer, especially with the detailed analysis of breast cancer patient P6, we decided that the main focus of the manuscript is based on breast cancer. Therefore, the data presented here have been pasted to the S2.
- Table 1:
- adding type of tumor to the top row of the table would increase the clarity
Thank you very much for this suggestion. We changed the table accordingly.
- add total number of CTC+ and CTC- pat in the top row for each tumor type
Thank you very much for this suggestion. We changed the table accordingly.
- total n for breast cancer is in the wrong line.
Thank you very much for this note, we corrected the position.
- for the genes, add the percentage of CTC+ cases for the wt
As the table is already very large we would like not to include it. For the reader it is rather easy to calculate it
- for the NSCLC group, mostly the absolute numbers and not % are given.
Thank you very much for this suggestion. We changed the table accordingly.
- in the NSCLC group/survival check the % of the dead patients (40% from 14 is not an even number)
The column indicates that 40% of the dead lung cancer patients were CTC positive.
- Figure 2 legend: remove some "-" within the word; revise "(B) CD74 and (C) CD44..."
We have now revised and clarified the figure legend for 2B and 2C. We also removed the additional “-“ from the legend.
- Fig 2D and E: total number of CTCs is 27, in the text it is 28
We apologize for the typing error.
- Fig 3 legend: (leukocyte exclusion marker: ")" is missing." (D) Copy number alteration profiles of single CTCs of P6 showing (a - omit) losses..."
Thank you very much for this note. We included “and CD45 as well” to the legend and removed the “-“.
Round 2
Reviewer 4 Report
The authors provided an improved version of the manuscripts.
However I would encourage you to:
- improve the quality of Fig 3D and E (barely legible)
- and to take another look to my previous comments to Table 1:
or the NSCLC group, mostly the absolute numbers and not % are given.
Thank you very much for this suggestion. We changed the table accordingly.
For EGFR still absolute numbers are given. Also 2 patients seem to be missing (5+1 is not 8)
in the NSCLC group/survival check the % of the dead patients (40% from 14 is not an even number)
The column indicates that 40% of the dead lung cancer patients were CTC positive.
Still, 40% of the dead lung cancer patients is not an integer. Please recheck.
Author Response
Dear Reviewer 4,
dear Editor,
thank you for your fast review and for your comments. We could address all raised points.
The authors provided an improved version of the manuscripts.
However I would encourage you to:
- improve the quality of Fig 3D and E (barely legible)
We are sorry that the quality of this particular figure was still not sufficient. We could improve the quality of Figure 3D and E, so that it’s well legible now.
- and to take another look to my previous comments to Table 1:
or the NSCLC group, mostly the absolute numbers and not % are given.
For EGFR still absolute numbers are given. Also 2 patients seem to be missing (5+1 is not 8)
in the NSCLC group/survival check the % of the dead patients (40% from 14 is not an even number)
The column indicates that 40% of the dead lung cancer patients were CTC positive.
Still, 40% of the dead lung cancer patients is not an integer. Please recheck.
We apologize for the incorrectness of the table. We carefully checked again the table and unfortunately there must have happened something while converting them, which we didn’t realize before. Some numbers changed by accident, we are very sorry for this. Now, the table is in its correct form.